# Providing Safe Space for Honest Mistakes in the Public Sector Is the Most Important Predictor for Work Engagement after Strategic Clarity

**Thais Gargantini [1], Michael Daly [2], Joseph Sherlock [2] and Teddy Lazebnik [3,*]**

1   Department of Psychology, Reichman University, Herzliya 4610101, Israel; thais.gargantinicard@post.idc.ac.il
2   Center for Advanced Hindsight, Duke University, Durham, NC 27708, USA; michael.daly@duke.edu (M.D.); jjs92@duke.edu (J.S.)
3   Department of Cancer Biology, Cancer Institute, University College London, London WC1E 7HU, UK
*   Correspondence: t.lazebnik@ucl.ac.uk; Tel.: +972-545524589

**Abstract:** Multiple studies highlight the link between engagement at work and performance, influencing organizations to put more effort into improving employee engagement levels. In this study, we begin to examine the influence of multiple psychological parameters on employees' work engagement (WE) within the public sector. The idea is to break the concept of WE down into eight individually measurable parameters that will allow for a better understanding and development of stronger interventions. Based on this analysis, we reproduce the outcome that strategic clarity is the most connected property to WE. More importantly, we introduce a new concept, honest mistakes, and show that having a safe space for making mistakes and learning from it is the second most important property of WE. This result is of interest, as allowing mistakes, even if they were made innocently, is considered taboo in the public sector. These outcomes are based on the reports of $n = 7682$ public sector employees from Brazil. In particular, the analysis shows that these outcomes hold for both professional and management positions across the health, administrative, justice, police, social work, and education offices.

**Keywords:** honest mistakes; public sector management; work engagement; dominance analysis; behavioral economics factors

## 1. Introduction

High levels of engagement at work (e.g., Work Engagement or *WE*) in the public sector directly impact the health, education, and economic services obtained by the population [1,2]. A large body of work has studied the connection between WE and productivity, showing a possible correlation between the two in a wide range of cultures, professions, and over time [3–13]. For instance, a two-year investigation by the National Health Service (NHS) in the United Kingdom (UK) showed that low WE in employees is linked to subsequent patient mortality, even when prior patient mortality is controlled for [14]. Additionally, WE is found to be positively correlated with nations' economic activity and productivity based on data of 43,850 employees from 35 European countries [15]. Similar dynamics occur in the private sector as well. Organizations with high levels of WE have significantly lower levels of workplace stress [16] and workplace accidents [17]. Engaged employees in private sector organizations have a higher perception of individual impact in addition to feeling more creative, innovative [18], and being physically healthier [19].

Nonetheless, some organizations are not putting sufficient emphasis on increasing employee WE [20,21]. The public sector, in particular, governments and their offices, are performing notably worse at engaging their employees than their private-sector counterparts [22]. This research examines the relative importance of money and eight

psychological parameters (strategic clarity, honest mistakes, work appreciation, caring environment, trust, clear expectations, psychological safety, autonomy) previously shown to influence WE levels.

While most studies about WE have focused on one specific mechanism and its influence on WE, this research examines the relative importance of money and additional eight psychological parameters (strategic clarity, honest mistakes, work appreciation, caring environment, trust, clear expectations, psychological safety, autonomy) previously shown to influence WE levels. In addition to showing the importance of having these parameters balanced in a work environment to achieve higher levels of WE, we introduces a new formalization of the Honest Mistakes concept, which is the perceived ability to make mistakes and learn/grow from them without facing significant repercussions.

A yet growing body of research has brought attention to how organizations should learn from mistakes, recognizing the importance of learning from small mistakes to avoid large ones [23–25]. However, its impact has been neglected in many other dimensions of the work environment. This research showed how this practice of having a safe space for honest mistakes has to be widespread across different domains, organizations and departments not only to avoid new mistakes but to foster productivity and WE.

The remainder of the paper is structured as follows: Section 2 outlines the conceptual framework we used. Section 3 presents our materials and methods. Section 4 lays out the empirical results of our study. Section 5 discusses the results and offers possible future work. Section 6 concludes the main outcome of the work and suggests an implementation method. A schematic view of the research structure is provided in Figure 1.

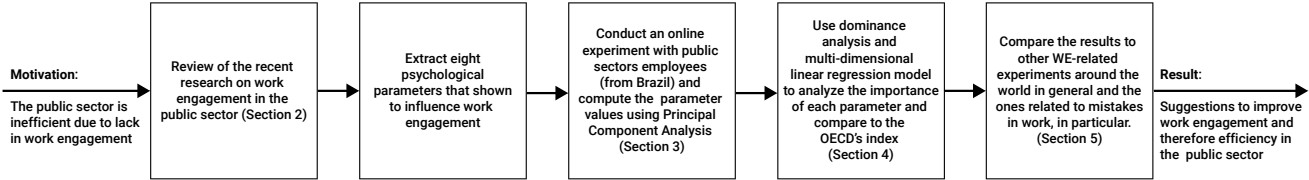

**Figure 1.** A schematic view of the research structure.

## 2. Background

Commonly, work engagement (WE) is defined as "a positive, fulfilling, work-related state of mind that is characterized by vigor, dedication, and absorption" [26]. WE is most often measured by the Utrecht WE index [27–29] alongside other scales [30–34]. The Organisation for Economic Co-operation and Development (OECD) adopted the Utrecht index to measure levels of WE in the public sector [35]. The adapted WE index has high validity and reliability. Nonetheless, this index does not help break down the social and psychological parameters that influence the WE in practice and simply provides an overview. Looking beyond the index, toward a more nuanced view of WE, would allow for the development of stronger interventions. Namely, the main contribution of this research is the breakdown of the WE into individually measurable parameters, providing a better understanding of the sociological and work environmental mechanisms that are taking part in defining the level of WE of employees.

The literature on WE and motivation has distinguished multiple mechanisms that have been shown to influence it empirically. Based on this literature of mechanisms and dependencies, we define eight parameters that show strong connections to WE across several cultures, sectors, and times: namely, strategic clarity, honest mistakes, work appreciation, a caring environment, trust, clear expectations, psychological safety, and autonomy. We also include monetary compensation in our analyses to be consistent with recent research. A schematic view of these parameters is provided in Figure 2, and a summary of these properties is provided in Table 1. Further descriptions of these parameters are found in the following paragraphs.

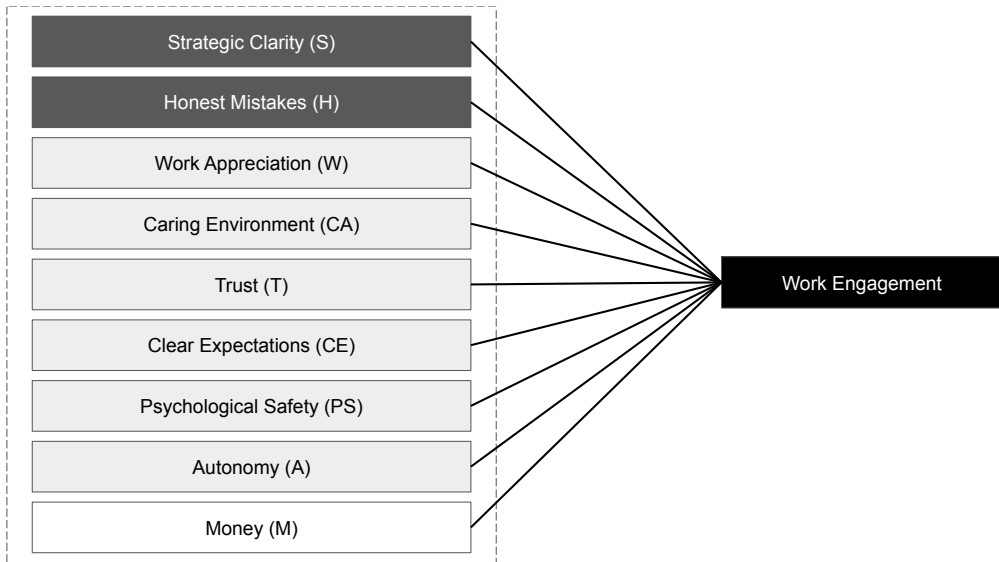

**Figure 2.** A schematic view of the psychological parameters that influence work engagement.

**Table 1.** A summary of the parameters used in the Work Engagement (WE) modeling.

| Name | Description |
|---|---|
| **Strategic clarity** | Feeling of purpose in one's work, alignment to company vision. |
| **Honest mistakes** | Perceived ability to make mistakes and learn/grow from them without facing significant repercussions. |
| **Work appreciation** | Continuous perception of organizational appreciation for one's individual contribution. |
| **Caring environment** | Willingness of coworkers to reciprocate care and consideration in social exchanges. |
| **Trust** | Trust in how one's organization and/or its leaders will behave in the future and transparency of policies and processes. |
| **Clear expectations** | Well-defined objectives and goals combined with well-given feedback. |
| **Psychological safety** | The absence of psychological and social risk or harm within a team, safety and support in taking risks. |
| **Autonomy** | Ability to exercise one's independent judgment at work, control over decisions within one's job. |
| **Money** | Absolute value of monetary compensation given to the employee as a result of the work, salary or in-kind compensation. |

Strategic clarity is related to meaning at work, i.e., purpose. The literature suggests that having a sense of purpose comes from two levels, finding intrinsic value in daily activities and believing the broader work to be worth doing, which is clearly the case for mission-oriented positions [36]. An extensive body of research has debated how much doing broadly meaningful work influences employee outcomes, but many believe meaningful work has significant positive impacts [37–39]. In addition, emphasizing meaning in daily activities has also been shown to increase motivation and productivity [40]. We primarily focus on the former of these components with the strategic clarity parameter, as this is more applicable at the organizational level.

Honest mistakes refers to how mistakes are perceived within an organization. The perception of mistakes at work, at the employee and team level, has been shown to be associated with work outcomes. Although it would seem intuitive for mistakes to impact outcomes such as productivity, Edmondson et al. found evidence supporting a relationship in the opposite direction when self-reported [41]. More productive teams were found to make more mistakes than unproductive teams, which qualitative data indicated stemmed

from a greater willingness to report mistakes rather than actually making more mistakes. However, a study on residents in the Netherlands found that those with burnout reported more mistakes than engaged ones [42]. The nuance with these results is attributed to how mistakes are perceived. According to past research, the challenge to learn from mistakes comes from the need to overcome a great psychological discomfort caused by the threat to one's self-esteem. This discomfort needs to be overcome to allow an investigative process to occur about the situation that leads to learning from it [41,43,44]. To do that, a positive culture around errors needs to be fostered. Recent studies suggest that a learning climate and mistake acceptance allow for true learning [45] and that there is a relationship between leaders' forgiveness and organizational performance [46]. Learning behavior also mediates team psychological safety and team performance [23,25]. Research suggests there is a fine line to walk with negative feedback, needing to create a slight sense of shame to motivate change but framed as a learning opportunity to support employees' recovery [47]. If organizations are not successful at limiting shame through the perception of mistakes, employee well-being is harmed in the short term and WE is harmed in the long term.

Caring environment is one that cares for its employees. Flourishing positive emotions of employees impacts levels of trust [48], influences how information is processed [49], builds enduring personal resources [50], and mediates the expression of values in behaviors [51,52]. Showing concern and respect for subordinates has a strong direct relationship with the degree to which employees are satisfied with their leaders [7] and their engagement at work [53].

Psychological safety is when there is a "shared belief held by members of a team that the team is safe for interpersonal risk-taking" [53,54]. Employees who feel high levels of psychological safety are more likely to be highly motivated. More impressively, a longitudinal study at Google found psychological safety to be the strongest predictor of highly successful teams [55]. One explanation for the importance of psychological safety within teams, in addition to its influence on WE, is its significant mediation of teams' creative output [56]. The growing research on psychological safety suggests seemly endless benefits toward employee outcomes, leading more researchers to focus on the drivers of psychological safety [57]. A popular variable influencing psychological safety is perceived organizational support, where high levels of support from leaders and managers result in high levels of psychological safety [58].

Work appreciation is about how employees feel valued at work. When leaders and teammates believe employees are capable and important and pass this information through daily behaviors and attitudes, employees will internalize such information and form positive self-evaluations, impacting their motivation [59,60].

Clear expectations is about the importance of employees knowing what is expected of them. Due to uncertainty aversion—a psychological factor that describes the tendency to prefer the known to the unknown—clarifying expectations is important to achieving high levels of WE. Setting clear goals and giving effective feedback positively impacts WE as well [7,61–63]. In addition, by informing and enabling personal improvement, evaluating one's deficiencies and focusing on positive change impacts WE [64].

Trust is a leading indicator of how employees believe an organization and/or its leaders will behave in the future [65]. Trust is correlated with engagement [66] and performance [52,67]. Management Transparency enables managers to set a personal example for their employees and establish an organizational culture of openness, trust, and sharing that encourages employees to take initiative and risk [68].

The ability to execute is when the employee knows the effort put into work is not going to be wasted. Successful managers understand that the real value of strategy can only be recognized through execution. As a recent survey of portfolio managers put it: "The ability to execute strategy was more important than the quality of the strategy itself". It doesn't matter how good the plan is if you cannot make it happen [69].

Autonomy refers to the degree of freedom and independence the employee has to exercise his/her judgment at work [70]. Perceived autonomy initiates regulatory processes

that are qualitatively different from those that are initiated when the functional significance of the events or context is controlled [71]. The degrees of freedom and independence that were given to the employee to exercise his/her judgment at work were found to enhance intrinsic motivation [72] and increased the ability to satisfy high needs such as a sense of meaning at work and a sense of self-realization, improving performance in the long term [73].

## 3. Materials and Methods

We conducted an online survey (the survey was carried out by Duke University in partnership with Lemann Foundation, Republica.org, Humanize and Brava Institute, and the National Council of Administration Secretaries) in which public-sector employees (from both the state and municipality level) answered 120 statements divided in 14 blocks. The first question asked participants if they were working remotely or in person. According to their answer, specific blocks about their perceptions about work, work role, psychological profile, and demographics were showed. The survey was written in Portuguese to make sure the participants fully understand the statements. The majority of statements used Likert scales between 1 and 5 for responses, with 1 standing for "totally disagree" and 5 standing for "totally agree". The participants did not obtain any payment for answering the survey and were asked to respond to every statement. The survey in both Portuguese and English is provided as a supplementary material.

Using the statements, we computed the eight psychological parameters using principal component analysis (PCA) [74], resulting in a single, complex variable that represents the aggregated answer of an individual. We then used these parameters to compute Pearson correlations and run a dominance analysis [75] to obtain the linear connection between the features and the impact of each on WE. Furthermore, we used a multi-dimensional linear regression model to compute a WE index based on the parameters and compared it with the OECD's WE index. To investigate differences between sectors and between professional and management positions, we used the relevant subset of data and repeated the analyses.

## 4. Results

The data from the survey were gathered between October (2020) and March (2021) from state public employees in Acre, Alagoas, Amapá, Ceará, Espírito Santo, Goiás, São Paulo, Santa Catarina, Sergipe and Tocantins. In addition, public employees from the municipality of Guarulhos in the State of São Paulo also participated in the survey. In total, $n_{total}$ = 16,654 participants took the survey. However, only $n$ = 7682 (46.13%) of the participants completed the survey in its entirety.

Participants' age ranged from 17 to 85 years with a mean of 43.14 and a standard deviation of 10.46. Substantially more females 60.78% responded to the survey than males 37.83%, whereas the remaining 1.38% of the participants preferred to either not declare their gender or declare themselves as non-binary. There also was a substantial skew toward one demographic: the race is divided into 62.1% Caucasian, 29.6% African, and 3.41% Hispanic. The remaining 4.88% of the participants were distributed between Native-Americans, Asians, and those who prefer not to declare their race.

Most participants' work in Sao Paulo (41.91%), followed by Espírito Santo (17.24%), Santa Catarina (14.91%), Tocantins (17.07%), Alagoas (6.62%), and Goias (2.02%). The remaining 3.23% of the participants are from 16 additional states in Brazil. Of the seven main professions in the public sector, 32.29% of participants are from the education sector, 21.2% are administrative workers, 11.18% are health workers, 9.16% are policemen, and the remaining 26.16% are divided between the justice sectors, social work, and cross-sectorial positions.

Using a multi-dimensional linear regression model, our data show that the average prediction based on the proposed eight psychological parameters (obtained with $R^2 = 0.83$) has a 0.55 ($p < 0.001$) linear correlation with the OECD's WE Index. In particular, the

Pearson correlation between SC and HM and the OECD WE Index is 0.45 ($p < 0.005$) and 0.48 ($p < 0.005$), respectively.

The full dominance analysis resulted in 0.086 and 0.078 for SC and HM, respectively, as shown in Table 2. The results are consistent when dividing the data for participants that hold professional and management positions with 0.068 and 0.093 for SC and 0.055 and 0.087 for HM, as summarized in Table 3. Similar results are obtained when one divides the data by sectors as described by Table 2.

The order of importance of the remaining six psychological parameters varied when dividing between professional and management positions and across sectors, as shown in both Tables 2 and 3. For instance, while for policemen, the Work Appreciation parameter is the third most important, for the justice and the administrative sectors, it is close to last. Similarly, the Trust parameter is the 3rd most important predictor for social service while being only the 5th most important for the education sector, policemen, and the justice sector. Money (i.e., the salary and any additional financial benefits employees get) is the least dominant parameter for seven of the nine (77.77%) sectors examined in our research.

**Table 2.** Dominance analysis, divided by field. The most and second most dominant parameters in each field are highlighted in bold.

|  | Justice (*n* = 117) | Police (*n* = 699) | Social Service (*n* = 100) | Administrative (*n* = 1617) | Education (*n* = 2463) | Health (*n* = 853) | All (*n* = 7682) |
|---|---|---|---|---|---|---|---|
| **Strategic clarity** | **0.140** | **0.069** | **0.092** | **0.093** | **0.060** | **0.103** | **0.086** |
| **Honest mistakes** | **0.130** | **0.101** | **0.078** | **0.077** | **0.049** | **0.104** | **0.078** |
| **Work appreciation** | 0.044 | 0.057 | 0.022 | 0.047 | 0.038 | 0.036 | 0.042 |
| **Trust** | 0.050 | 0.035 | 0.066 | 0.037 | 0.025 | 0.042 | 0.037 |
| **Caring environment** | 0.080 | 0.040 | 0.026 | 0.031 | 0.026 | 0.047 | 0.034 |
| **Psychological safety** | 0.032 | 0.016 | 0.018 | 0.027 | 0.019 | 0.039 | 0.024 |
| **Clear expectations** | 0.040 | 0.032 | 0.041 | 0.027 | 0.017 | 0.033 | 0.023 |
| **Autonomy** | 0.016 | 0.006 | 0.017 | 0.011 | 0.021 | 0.026 | 0.023 |
| **Money** | 0.051 | 0.001 | 0.032 | 0.006 | 0.005 | 0.025 | 0.002 |

**Table 3.** Dominance analysis, divided into management and professional positions.

|  | Management Position (*n* = 2213) | Professional Position (*n* = 5250) |
|---|---|---|
| **Strategic clarity** | **0.068** | **0.093** |
| **Honest mistakes** | **0.055** | **0.087** |
| **Work appreciation** | 0.038 | 0.042 |
| **Trust** | 0.030 | 0.039 |
| **Caring environment** | 0.037 | 0.033 |
| **Psychological safety** | 0.020 | 0.024 |
| **Clear expectations** | 0.024 | 0.022 |
| **Autonomy** | 0.016 | 0.023 |
| **Money** | 0.002 | 0.002 |

## 5. Discussion

We investigated the influence of strategic clarity (SC), safe space for honest mistakes (HM), work appreciation, caring environment, trust, clear expectations, psychological

safety, autonomy, and money on work engagement (WE) among public sector employees. Based on data from $n = 7682$ government employees in Brazil, we show that financial compensation (e.g., money) has little connection to WE. This result supports multiple studies suggesting that appropriate financial compensation in the eyes of an employee is a necessary condition for WE but not enough to increase it above a minimal level [76–79].

Our analysis suggests that SC is the most important parameter for WE, reproducing results from previous researchers [80,81]. We show that employees perceiving a safe space for honest mistakes is the second most important parameter for WE, in general, and the most important parameter (over SC) for 22% of the sectors analyzed.

Although there is a slight distinction between these two parameters in terms of their impact on WE, there is a large difference in how much organizations focus on them. Organizations across industries attempt to motivate employees by giving them a sense of Strategic Clarity; however, only a few industries, such as technology and innovation, foster a learning culture within their teams. For many organizations, especially in the public sector, holding a safe space for honest mistakes is considered taboo.

A large body of work supports the importance of Honest Mistakes, showing that individuals require a safe space for exploration to make errors and learn from them, both personally and professionally [41]. This evidence contrasts with current practice in the public sector, where there is a low tolerance for mistakes in operations and significant negative feedback is directed at employees when mistakes occur. A more unique challenge to this sector also exists where there is negative feedback originating outside of the organization, as mistakes are more likely to become public knowledge.

In addition to limiting the efficiency of public sector organizations by depressed work engagement, learning, and creativity, not holding a safe space for honest mistakes limits the sharing of tacit knowledge. Tacit knowledge is held by individuals based on their accumulation of experiences within their role and organization [82]. Research indicates that while organizations have largely maximized the spread of easily-documented explicit knowledge, there is significant room for growth in performance by being more attentive to tacit knowledge [83]. The culture of sharing developed by holding a safe space for honest mistakes helps teams learn more from mistakes and other unique experiences that are less likely to be discussed [84].

If corrected, public sector organizations could properly foster employees' personal growth and improve work engagement, in turn yielding increases to both individual-level productivity and organizational performance. As far as we know, we are the first to highlight this outcome in the public sector in general and for the Brazilian public sector.

Our analysis is based on the data from a single country (Brazil) which may reflect a political or cultural bias in our results. Furthermore, we rely solely on self-reported survey data—within which people may knowingly or not misrepresent their perspective. Finally, the study ran during a period of general public frustration at perceived mistakes made around governance of the COVID-19 outbreak in Brazil, which we speculate may have heightened the public awareness of mistakes as well as how important they are perceived to be. These factors all suggest that further research across cultural and temporal contexts and with a variety of data collection approaches is needed to shed more light on the robustness of our results. Nevertheless, this the analysis presented here offers a notable contribution in the development of this theory.

Although the data for this research come from one particular country, the eight psychological parameters were built considering empirical evidence found in several parts of the world and in different contexts. Our hypothesis is that these parameters have a great influence on levels of WE in any other settings and countries. As for the order of importance, we could expect more variability. In our data, Strategic Clarity and Honest Mistakes were shown to have high importance across the different subsets, while the other six parameters were less consistent. We hypothesize that the order of importance of those parameters would have greater variability due to cultural differences.

The survey conducted for this research could be improved by reducing the number of questions proposed. From the $n_{total}$ = 16,654 participants that took the survey, $n$ = 7682 (46.13%) of them completed in its entirety. With greater knowledge of how to build the psychological parameters, the number of the questions can be largely reduced along with the estimated time to answer the survey in its entirety, diminishing the likelihood that participants will drop out before the end.

## 6. Conclusions

According to a recent meta-analysis across the public, semi-public and private sectors, the psychological construct WE received little attention within semi-public and public sectors [85]. However, the same research has shown that engaged public employees are more satisfied and committed to their jobs than semi-public and private sector employees. The government should put the effort into increasing levels of WE in the public sector. As an important effort in this direction, public sector offices need to accept that employees should make mistakes and need to prepare organizational structures accordingly to maximize work engagement. They also ought to develop feedback mechanisms that support employees' learning rather than emphasizing negativity and criticism.

Given the recommendation above, further research is suggested to better understand the causal relationship between the proposed psychological parameters and WE. Furthermore, exploring, experimentally, what is the right way to admit a mistake in order to increase the learning process and perception of safety could add relevant information to this debate.

One important way to further investigate and better validate the proposed conclusions is to conduct an empirical experiment in a public-sector organization implementing interventions that aim to influence positively one or more of the eight psychological parameters presented in this work. The outcome of such experiment would shed light on the sensitivity of the WE level to each parameter and how hard it is for public-sector organizations to influence it, which is one of the main limitations of this study.

**Author Contributions:** Conceptualization, T.G.; methodology, T.G., M.D., J.S. and T.L.; software, T.G. and T.L.; validation, M.D.; formal analysis, T.G. and T.L; investigation, T.G. and M.D.; resources, M.D. and J.S.; data curation, T.G., M.D. and J.S.; writing—original draft preparation, T.G. and T.L; writing—review and editing, T.G., M.D. and T.L.; visualization, T.G.; supervision, T.L.; project administration, T.G.; funding acquisition, T.G. All authors have read and agreed to the published version of the manuscript.

**Funding:** This research received no external funding.

**Institutional Review Board Statement:** Project Title: "D0790 Behavioral Economics and Municipal Policy". IRB Protocol Number: "2017-0506, Amendment 23". Researcher(s): Dan Ariely, Mariel Beasley, Joseph Sherlock, Lyndsay Gavin, Michael Daly, Jet Sanders, Nina Bartmann, Caroline Olsen, Min Lee, Dan Rosica, Anson Tong, Thais Gargantini Cardarelli, Aasha Reddy, Submission Date: 2 January 2021.

**Informed Consent Statement:** Informed consent was obtained from all individual participants included in the study.

**Data Availability Statement:** All the data used in this research are available upon written request from the authors.

**Acknowledgments:** The authors would like to thank the individuals from the Lemann Foundation, República.org, Brava Foundation, Humanize Institute, the Center for Advanced Hindsight at Duke University, Kayma, the National Council of Administration Secretaries, the State Governments from Acre, Alagoas, Amapá, Ceará, Espírito Santo, Goiás, São Paulo, Santa Catarina, Sergipe, Tocantins and the municipality of Guarulhos in the State of São Paulo in Brazil for there help in the data collection, funding, and refining the research question. In addition, the authors would like to thank Tali Regev and Elizaveta Savchenko for critically reviewing the paper.

**Conflicts of Interest:** The authors declare no conflict of interest.

**Abbreviations**

The following abbreviations are used in this manuscript:

| | |
|---|---|
| WE | work engagement |
| PCA | principal component analysis |
| OECD | Organisation for Economic Co-operation and Development |
| SC | strategic clarity |
| HM | honest mistakes |

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
