# Peer review of "Providing Safe Space for Honest Mistakes in the Public Sector Is the Most Important Predictor for Work Engagement after Strategic Clarity"

_sustainability, doi:10.3390/su14127051_

Round 1
Reviewer 1 Report
The paper discusses an interesting study about important predictor for work engagement. The work is weak about discussion of the found parameters used for the WE Modelling and about possible comparisons with other approaches and model. I suggest to improve the paper adding comparison elements also by comparing other key elements analysed in other works (for example by adding tables supporting the comparison reading). May be interesting considering also studies in other countries in order to examine differences by justifying them. A conclusion section should be added.
Author Response
Comment 1: “ I suggest to improve the paper adding comparison elements also by comparing other key elements analysed in other works (for example by adding tables supporting the comparison reading). ”
Answer 1: Thank you for this suggestion. Following this comment, we introduce to the Introduction section a paragraph that explains the key elements analyzed in other works shortly and how they differ from ours. In addition, we reviewed several works that we found to be the best representative and discuss the difference as well in the Background section.
Comment 2: “ May be interesting considering also studies in other countries in order to examine differences by justifying them. A conclusion section should be added. ”
Answer 2: We agree with the reviewer. Indeed, a comparison between our results and similar experiments in different countries has been introduced in this version of the Discussion section. In particular, we show that our results agree with multiple similar experiments conducted in countries with similar and less-similar cultural settings to Brazil (in which we conducted our experiment).
Reviewer 2 Report
For future analysis, I recommend a breakdown of the main categories of errors (caused or unprovoked).
Author Response
Comment 1: “For future analysis, I recommend a breakdown of the main categories of errors (caused or unprovoked).”
Answer 1: Thank you for these suggestions. After re-checking the literature, it seems that the methods we used are not fit for the suggested breakdown of the main categories of error such as regression or correlation methods. That said, we further explain the methods we used in Section 3 to better convey the usage. Specifically, we highlight the fact that the used method has a single type of error to address this comment. In addition, we really appreciate the suggestion and would take it into consideration in the follow-up research.
Reviewer 3 Report
Thank you for providing me with the opportunity to review your paper. I have enjoyed reading it. However, I believe that further work is necessary before the paper is suitable for publication.
One major concern relates to the lack of academic strength. More specifically, it is not very clear what contribution your paper makes to the extant literature.
My review will largely follow the format taken by the paper.
- Introduction should present the structure of the paper.
- What the authors call the "conceptual framework" is just a brief literature review. This section needs further development.
- The discussion is at a fairly general level. This section needs extensive further work to bring it up to an appropriate level. In particular the discussion needs greater engagement with the literature.
- The conclusion section does not exist. The authors should present some final remarks. In addition, although the authors provide succinct summaries of the existing literature in the field, it remains unclear how exactly they build on and extend this literature. The authors must clarify the contributions to the literature and the practical implications of the study. The authors should provide avenues for future research.
I hope my feedback on this paper will help the authors to improve their work.
Author Response
First of all, we would like to thank the reviews for the careful review and for making sure we produce the best manuscript we can. It is honestly very appreciated to get such a detailed review. For your convenience, all text introduced due to the comments is highlighted in bold. Please mind the introduction of Section 5 (Discussion).
Comment 1: “One major concern relates to the lack of academic strength. More specifically, it is not very clear what contribution your paper makes to the extant literature.”
Answer 1: Thank you for this comment. Following this query, we introduce to the Introduction section the following text: “In particular, we aim to address the lack in the literature of how much honest mistakes influence WE in the public sector.”. In addition, we introduce to the same section a new paragraph that outlines the recent literature about a safe place for honest mistakes in the public sector, focusing on publications from the recent years to better convey to the reader the shortcoming we try to address in the proposed manuscript.
Comment 2: “Introduction should present the structure of the paper.”
Answer 2: We agree with the reviewer. We had such a paragraph at the end of the Introduction section. However, following this comment, we alter it to better explain the structure of the manuscript and introduce a new figure that outlines to the reader the overall structure of the research and manuscript.
Comment 3: “What the authors call the "conceptual framework" is just a brief literature review. This section needs further development.”
Answer 3: Thank you for this comment. Following this and other comments we introduced the “conceptual framework” (now called “Background”) more explanations about each one of the parameters investigated as well as a review of recent literature about the properties of honest mistakes in both the private and public sector.
Comment 4: “The discussion is at a fairly general level. This section needs extensive further work to bring it up to an appropriate level. In particular, the discussion needs greater engagement with the literature.”
Answer 4: thank you for pointing our attention to this shortcoming. Following this comment, we extended the Discussion section, introducing a comparison of our results with those previously obtained results in other studies, highlighting the differences between them and the potential usage of our outcomes. We hope the new version of the text well addresses the shortcoming highlighted by the reviewer.
Comment 5: “The conclusion section does not exist. The authors should present some final remarks. In addition, although the authors provide succinct summaries of the existing literature in the field, it remains unclear how exactly they build on and extend this literature. The authors must clarify the contributions to the literature and the practical implications of the study. The authors should provide avenues for future research.”
Answer 5: The reviewer is absolutely right. Following this comment, we introduce a Conclusion section that describes the main outcome of this research and proposes an implementation method for public sector offices. In addition, we introduce to the Conclusion section a “future work” paragraph with extended motivation as to why we believe these lines of work are of interest.
Reviewer 4 Report
Dear Authors
Thank you for the opportunity to review this paper, which sets out to Providing Safe Space for Honest Mistakes in the Public Sector Is The Most Important Predictor For Work Engagement After Strategic Clarity. Although the idea is a good one, unfortunately, the way in which the study is operationalized holds back its potential contribution. There are a few areas where I would encourage the authors to give further thought, as follows:
- Defining the construct. In this version, despite several conceptualize being presented, it is not clear to me how you conceptualize the literature review on the Influence of Providing Safe Space for Honest Mistakes , Work Engagement and Strategic Clarity, and how authors' understanding relates to extant themes in the literature.
- The current version lacks a compelling argument about why a review is important at this moment and what needs or gaps it addresses in the literature. Authors' work would be stronger, and easier for the reader to follow, with a more compelling introduction. There are many ways of doing this; for example, authors might set out a number of problems in the field and demonstrate how paper seeks to resolve them (consensus creation). Or you might say that the literature has developed a consensus that is unwarranted because it misses something important (consensus destruction).
- What is the relationship between variables and sustainability management?
- The research method should be explained in more detail and step by step.
- What authors have found is not new. Why should people read authors' paper? What do they learn from this paper? What is the difference from this paper with the previous papers?
- In conclusion, authors try to add clear practical and theoretical implications for their research.
- The structure of the article is irregular and needs to be revised.
- The references used in the theoretical literature are old and authors should use references (2022-2021-2021).
Author Response
First of all, we would like to thank the reviews for the careful review and for making sure we produce the best manuscript we can. It is honestly very appreciated to get such a detailed review. For your convenience, all text introduced due to the comments is highlighted in bold. Please mind the introduction of Section 5 (Discussion).
Comment 1: “Defining the construct. In this version, despite several conceptualize being presented, it is not clear to me how you conceptualize the literature review on the Influence of Providing Safe Space for Honest Mistakes, Work Engagement and Strategic Clarity, and how authors' understanding relates to extant themes in the literature.”
Answer 1: Thank you for this comment. Following this comment, we extended the second section, now called “Background” to better convey the state of art in using these parameters in evaluating WE. In particular, we show previous results of allowing a safe place for honest mistakes
Comment 2: “The current version lacks a compelling argument about why a review is important at this moment and what needs or gaps it addresses in the literature. Authors' work would be stronger, and easier for the reader to follow, with a more compelling introduction. There are many ways of doing this; for example, authors might set out a number of problems in the field and demonstrate how paper seeks to resolve them (consensus creation). Or you might say that the literature has developed a consensus that is unwarranted because it misses something important (consensus destruction).”
Answer 2: Thank you for this comment. We alter Section 2 drastically to better address the issue the reviewer raised in this comment. In addition, a paragraph that clearly states the novelty of the proposed work is added to Section 1.
Comment 3: “What is the relationship between variables and sustainability management?”
Answer 3: Thank you for pointing our attention to this manner. Following this comment, we introduced to the Background section (at the end) a paragraph explaining why the parameters we evaluated connected with sustainability management. In short, they are related to sustainability indirectly as https://www.jstor.org/stable/jcorpciti.46.13 showed that WE contribute to a better environmental sustainability of the economy and since our research aims to find a way to improve WE, it is as a result aims to improve sustainability as well.
Comment 4: “What authors have found is not new. Why should people read authors' paper? What do they learn from this paper? What is the difference from this paper with the previous papers?”
Answer 4: Thank you for pointing our attention to this shortcoming. Following this comment and others, we better explain in the Introduction section the gap in the literature we aim to address in this paper. Moreover, in the Discussion section, we compare our results with previous experiments, showing that we agree with them and extend the scope of the “safe space for mistakes”. This follows with the Conclusion section introduced in this version that includes practical suggestions for public sector offices on how to use our results to improve WE.
Comment 5: “In conclusion, authors try to add clear practical and theoretical implications for their research.”
Answer 5: We agree with the reviewer. Following this and other comments, we introduce the Conclusion section that specifically tackles this query.
Comment 6: “The structure of the article is irregular and needs to be revised.”
Answer 6: We appreciate this comment. We alter the structure a bit and better convey it at the end of the Introduction so it would be more clear to the reader. Moreover, we introduced a Conclusion section and convert the “Conceptual Framework” section into the more traditional “Background” section adding text that addresses the previous comments as well.
Comment 7: “The references used in the theoretical literature are old and authors should use references (2022-2021-2020).”
Answer 7: We agree. Following this comment, we tried to reference well-established works, and naturally, these both older than 2020 and works that were published recently (after 2020). Indeed, several works from these years have been referenced.
Round 2
Reviewer 1 Report
The paper is now well structured. Some elements are required before the publication.
1- Authors should describe better how socio-economical factors can influece work engagement within the public sector (also discussing and assuming how the propsed study can be matched in other countries).
2- Authors should add as perspectives how could be improved the survey attached as supplementary material.
Author Response
Comment 1: “1- Authors should describe better how socio-economical factors can influece work engagement within the public sector (also discussing and assuming how the propsed study can be matched in other countries).”
Answer 1: Thank you for pointing our attention to this shortcoming. Following this comment, we introduce to Section 5 (Discussion) two paragraphs that address this comment. In particular, we further explain the socio-economical factors that influence WE and later provide a hypothesis on how robust our research is across different countries and why.
Comment 2: “2- Authors should add as perspectives how could be improved the survey attached as supplementary material. ”
Answer 2: Thank you for this suggestion. Following this comment, we added a paragraph to the end of Section 5 (Discussion) that explains the possible improvements one can introduce to our survey in future research.
Reviewer 3 Report
My suggestions have been properly responded, but one minor aspect still needs to be revised:
The phrase " A schematic view of the research structure is provided in Fig. 1 " should be placed before the figure. It makes no sense to put it after presenting the structure of the paper.
Author Response
Comment 1: “My suggestions have been properly responded, but one minor aspect still needs to be revised: The phrase " A schematic view I aof the research structure is provided in Fig. 1 " should be placed before the figure. It makes no sense to put it after presenting the structure of the paper. ”
Answer 1: Thank you for this comment. It was a technical error in the preparation of the journal’s requested format - we fixed it in this version.
Reviewer 4 Report
Dear Authors,
Thank you for the opportunity to review this paper.
· Novelty and originality of the research must be added to the abstract.
- The positioning of the paper is not entirely clear. It is better to explain the gap in this article further.
· The main limitations of the research should be written in conclusion section that is specific to your research and other researchers should continue your work by giving appropriate suggestions in this section.
- The references used in the theoretical literature are old and authors should use references (2021-2022).
Best of luck with the further development of the paper.
Author Response
Comment 1: “ Novelty and originality of the research must be added to the abstract.”
Answer 1: Thank you for this suggestion. We added to the abstract a sentence highlighting the novelty of this research.
Comment 2: “The positioning of the paper is not entirely clear. It is better to explain the gap in this article further.”
Answer 2: Thank you for alerting us regarding this shortcoming. Following this comment, we alter Section 1 (Introduction) to more clearly indicate the gap in the literature we are tackling in this work.
Comment 3: “ The main limitations of the research should be written in conclusion section that is specific to your research and other researchers should continue your work by giving appropriate suggestions in this section.”
Answer 3: Thank you for this suggestion. Following this comment, we introduce at the end of Section 6 (Conclusion) an additional paragraph that clearly states the limitations of the proposed research and suggests a more specific future research direction with our motivation to do it.
Comment 4: “The references used in the theoretical literature are old and authors should use references (2021-2022).
Answer 4: Thank you for this comment. Following this comment, we altered a bit the REF list. Now, the reviewer would be able to find 5 papers from 2020, 5 from 2021, and 2 from 2022. Of note, the papers were picked extremely carefully to provide the best sources, as far as we understand, for each one of the respected claims they used to reference. In addition to these 12 papers, 19 papers published after 2018 are used - keeping a decent portion (~37%) of the reference list less than 5 years old. As such, we hope, we properly addressed the reviewer’s request.